# Boosting Summarization with Normalizing Flows and Aggressive Training

**Yu Yang**
University of Minnesota
yang6367@umn.edu

**Xiaotong Shen**
University of Minnesota
xshen@umn.edu

## Abstract

This paper presents FlowSUM, a normalizing flows-based variational encoder-decoder framework for Transformer-based summarization. Our approach tackles two primary challenges in variational summarization: insufficient semantic information in latent representations and posterior collapse during training. To address these challenges, we employ normalizing flows to enable flexible latent posterior modeling, and we propose a controlled alternate aggressive training (CAAT) strategy with an improved gate mechanism. Experimental results show that FlowSUM significantly enhances the quality of generated summaries and unleashes the potential for knowledge distillation with minimal impact on inference time. Furthermore, we investigate the issue of posterior collapse in normalizing flows and analyze how the summary quality is affected by the training strategy, gate initialization, and the type and number of normalizing flows used, offering valuable insights for future research.

## 1 Introduction

Abstractive summarization (See et al., 2017; Paulus et al., 2018; Wang et al., 2018) aims to generate summaries by rephrasing or introducing novel words to capture the most salient information in the source text. Many abstractive summarization models (Liu and Lapata, 2019; Zhang et al., 2020a; Rothe et al., 2020; Raffel et al., 2020) are based on the Transformers architecture (Vaswani et al., 2017) and have consistently produced state-of-the-art summarization quality. However, issues such as exposure bias (Ranzato et al., 2016; Qi et al., 2020), lack of text generation diversity (Holtzman et al., 2020), and insufficient capturing of semantic information (Reimers and Gurevych, 2019; Wang et al., 2020) remain.

Variational models have gained increasing research interest (Zhang et al., 2016; Su et al., 2018; Wang et al., 2019; Fu et al., 2020) as they address these issues by introducing uncertainty in predictions through learning a probability distribution over latent variables. A variational model enables diverse text generation (Du et al., 2022), smoother output spaces, and semantically meaningful latent codes (Wang et al., 2019) that guide the generation of coherent and informative summaries.

Nonetheless, existing variational models have not fully achieved the aforementioned desirable properties due to two main challenges. Firstly, the semantic information in the source text may possess a complex structure. However, since introducing latent variables complicates parameter estimation, many current models (Fu et al., 2020; Zheng et al., 2020) represent latent codes using a Gaussian distribution, which is insufficient for capturing the intricacies of the latent space and could potentially reduce model performance. To enrich latent distributions, researchers suggest replacing the highly restricted isotropic Gaussian with normalizing flows (Rezende and Mohamed, 2015). Normalizing flows can generate complex distributions while preserving density in an analytical form, and they have been integrated into variational autoencoder (VAE) (Kingma and Welling, 2014; Rezende et al., 2014) and variational encoder-decoder (VED) (Serban et al., 2017; Zhou and Neubig, 2017) frameworks to better approximate the latent posterior. This approach has found application in various domains, including text generation (Wang et al., 2019), neural machine translation (Setiawan et al., 2020), and dialogue generation (Luo and Chien, 2021). Despite this progress, the operating characteristics of normalizing flows on summarization tasks have yet to be investigated.

Secondly, as reported by previous studies (Bowman et al., 2016; Kingma et al., 2016; Chen et al., 2017), variational models tend to experience posterior collapse during training, which occurs when the KL term vanishes to zero, indicating that the model fails to learn meaningful latent codes. This

problem becomes more severe when modeling discrete data with a strong auto-regressive decoder (He et al., 2019), which is the case for Transformer-based summarization models. To resolve this issue, several solutions have been proposed, such as employing a less auto-regressive decoder network (Yang et al., 2017; Semeniuta et al., 2017; Shen et al., 2018a), modifying the training objective (Zhao et al., 2017; Tolstikhin et al., 2018; Prokhorov et al., 2019), and proposing new training strategies (Kim et al., 2018; He et al., 2019). However, most existing work focuses on the VAE framework with Gaussian latent distribution, yet limited work considers the VED framework with normalizing flows. In particular, two questions remain unclear: (1) when the latent distribution is modeled by normalizing flows, does the posterior collapse problem still exist? (2) when posterior collapse exists, what are the appropriate strategies to achieve good summarization quality within the VED framework?

This paper introduces FlowSUM[1], a normalizing flows-based VED framework for Transformer-based summarization, along with a controlled alternate aggressive training (CAAT) strategy and a refined gate mechanism to resolve the two challenging issues. Our contributions include:

1. We employ normalizing flows to enrich the latent posterior distribution and integrate the latent code into Transformer-based models in a plug-and-play manner, demonstrating its effectiveness through extensive experiments.

2. We propose a controlled alternate aggressive training strategy and a refined gate mechanism to mitigate the posterior collapse problem and improve training efficacy.

3. Our findings suggest that FlowSUM facilitates knowledge distillation while having a negligible effect on inference time, implying normalizing flows' potential for transferring knowledge from advanced large language models.

4. We investigate the posterior collapse problem for different normalizing flows and examine how the quality of a summary is impacted by the training strategy, gate initialization, and the type and depth of normalizing flows.

---

[1]Code is available at `https://github.com/yuyangstat/flowsum`.

This article consists of five sections. Section 2 provides an overview of normalizing flows, VED, and a summary of related studies. Section 3 describes the proposed model architecture and the training strategies employed. Section 4 presents the experimental setup and results, and Section 5 concludes the paper with some discussions.

## 2   Backgrounds

### 2.1   Normalizing Flows

Normalizing flows (NF) (Rezende and Mohamed, 2015) is a type of generative model that has gained popularity in recent years. The fundamental idea involves mapping a simple probability density (e.g., Gaussian) to a more complex one through a series of invertible transformations. One of the key advantages of NF is that it allows for exact likelihood evaluations, which is crucial for many applications such as density estimation (Papamakarios et al., 2017), data generation (Tran et al., 2019), and variational inference (Kingma et al., 2016). A flow-based model consists of two components: a base distribution $p_{\mathrm{u}}(\mathbf{u})$ and a transformation $f(\cdot) : \mathbb{R}^D \to \mathbb{R}^D$, where $f$ must be invertible and both $f$ and $f^{-1}$ must be differentiable. Let $\mathbf{x} = f(\mathbf{u})$ where $\mathbf{u} \sim p_{\mathrm{u}}(\mathbf{u})$, then the density of $\mathbf{x}$ can be obtained via a change of variables (Bogachev, 2007):

$$
\begin{aligned}
p_{\mathrm{x}}(\mathbf{x}) &= p_{\mathrm{u}}(\mathbf{u}) \left| \det J_f(\mathbf{u}) \right|^{-1} \\
&= p_{\mathrm{u}}(f^{-1}(\mathbf{x})) \left| \det J_{f^{-1}}(\mathbf{x}) \right|.
\end{aligned} \tag{1}
$$

In this paper, we examine several NFs, including planar flows (Rezende and Mohamed, 2015), radial flows (Rezende and Mohamed, 2015), Sylvester flows (van den Berg et al., 2018), real-valued non-volume preserving (RealNVP) transformation (Dinh et al., 2017), inverse autoregressive flow (IAF) (Kingma et al., 2016), rational-quadratic neural spline flows (RQNSF) (Durkan et al., 2019), and rational-linear neural spline flows (RLNSF) (Dolatabadi et al., 2020). We delegate the detailed discussion of transformation and invertibility to Appendix J. Throughout the paper, for each type, we compose $K$ layers of transformation $f_K \circ \cdots \circ f_1(\cdot)$, which remains invertible and differentiable.

### 2.2   Variational Encoder-Decoders

Variational encoder-decoders (VEDs) (Zhang et al., 2016; Serban et al., 2017; Zhou and Neubig, 2017; Shen et al., 2018b), which can be seen as an extension of variational autoencoders (VAEs) (Kingma

and Welling, 2014; Rezende et al., 2014), have been widely used to understand the conditional data generation process. Given an input $x$, the framework posits the existence of a latent variable $z \sim p(z \mid x; \phi)$, and the generation of $y$ relies on $p(y|x, z; \theta)$. With this premise, the conditional data generation can be formulated as in Eq. 2.

$$p(y \mid x; \phi, \theta) = \int p(z \mid x; \phi)p(y \mid x, z; \theta)dz \tag{2}$$

Since the marginal $p(y \mid x; \phi, \theta)$ is intractable, we employ variational inference to estimate the parameters. This involves maximizing the evidence lower bound (ELBO), a surrogate of the log-likelihood, as defined in Eq. 3. The underlying idea is to propose a parameterized distribution $q(z \mid x, y; \psi)$, known as the variational posterior, to approximate the true posterior distribution $p(z \mid x, y; \phi, \theta)$. The greater the flexibility in $q(z \mid x, y; \psi)$, the better the approximation, and the more effective the surrogate ELBO becomes. See more details in Appendix B.

$$\begin{aligned}
&\text{ELBO}_{\text{VED}} \\
&= \mathop{\mathbb{E}}_{q(z|x,y;\psi)}[\log p(y \mid x, z; \theta)] - \text{KL}(q(z \mid x, y; \psi)\|p(z \mid x; \phi))
\end{aligned} \tag{3}$$

For summarization, we parameterize $p(y \mid x, z; \theta)$ as an encoder-decoder model that generates summaries conditioned on the input text and latent code.

## 2.3 Related Work

### 2.3.1 Transformer-based Summarization Models

Transformer-based models equipped with pre-training and fine-tuning techniques have enjoyed significant success in many NLP tasks, including text summarization. Liu and Lapata (2019) proposed BertSUM for extractive and abstractive tasks, utilizing the pre-trained BERT encoder (Devlin et al., 2019). To better align the pre-trained encoder for document understanding with the decoder trained from scratch for text generation, Rothe et al. (2020) demonstrated the effectiveness of leveraging pre-trained BERT (Devlin et al., 2019), GPT-2 (Radford et al., 2019), and RoBERTa (Liu et al., 2019) checkpoints to build sequence-to-sequence (S2S) models for tasks including summarization. Another approach is to address both document understanding and generation in a unified framework by first pre-training some general-purpose S2S models and then fine-tuning on downstream tasks,

for instance, BART (Lewis et al., 2020), MASS (Song et al., 2019), UniLM (Dong et al., 2019), ProphetNet (Qi et al., 2020), and T5 (Raffel et al., 2020). In addition, Zhang et al. (2020a) proposed PEGASUS with a pre-training objective tailored for abstractive summarization, achieving significant improvements across multiple datasets.

### 2.3.2 Variational Summarization

Variational summarization models come in two different flavors: unsupervised and supervised. In the unsupervised domain, researchers commonly utilize variational autoencoders in conjunction with specific control mechanisms for summary generation, as exemplified by prior work such as Schumann (2018); Chu and Liu (2019); Brazinskas et al. (2020). In the supervised realm, there are generally two primary approaches. The first approach models the conditional probability of the target sentences $p(y \mid x)$ as in Eq. 2, whereas the second approach models the joint probability of the source and target sentences $p(x, y)$ with $\int p(z)p(x \mid z)p(y \mid z, x)dz$. Our model belongs to the first category, akin to prior studies like Setiawan et al. (2020); Fu et al. (2020). In contrast, other works, including Zheng et al. (2020); Nguyen et al. (2021); Zou et al. (2021), adopt the second type by jointly modeling topics and sequence-to-sequence generation. Most of them assume a simple Gaussian latent prior, except for Nguyen et al. (2021), which employs normalizing flows to model neural topic models and enrich global semantics. However, they did not specify the choice of normalizing flows and how they addressed posterior collapse. To the best of our knowledge, there remains limited research on the application of normalizing flows in variational summarization models and their operating characteristics.

## 3 Normalizing Flows Enhanced Summarization Model

### 3.1 FlowSUM Model Architecture

As illustrated in Fig. 1, FlowSUM consists of three components: an NF latent module, a Transformer-based encoder-decoder, and a refined gate mechanism. The NF latent module focuses on modeling the variational posterior $q(z \mid x, y; \psi)$, whereas the encoder-decoder, combined with the refined gate, models the conditional generation $p(y|x, z; \theta)$ with latent code. As a simplification, we assume the conditional prior $p(z \mid x; \phi)$ is a standard Gaussian as

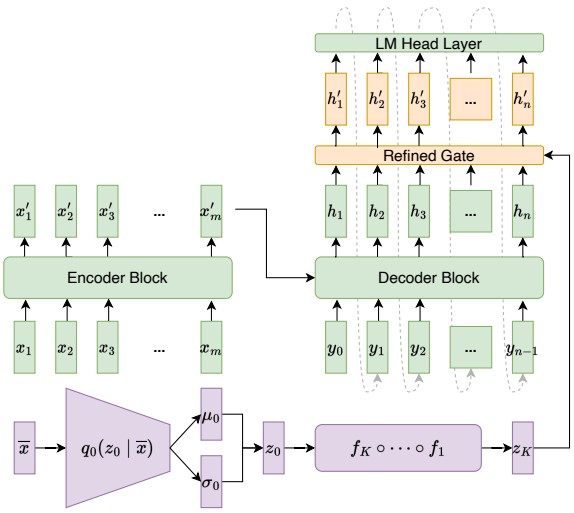

Figure 1: FlowSUM Model Architecture, including an NF latent module (in purple), a Transformer-based encoder-decoder (in green), and a refined gate mechanism (in orange)

in Setiawan et al. (2020). Throughout this section, let $e$ be the embedding size, $m, n$ be the length of the input source and target summary respectively, $\ell$ be the latent dimension of the NF latent module, $d$ be the dimension of the decoder's hidden states, $\{x_i\}_{i=1}^m$ be the input source text, $\{y_j\}_{j=1}^n$ be the target summary text, and $\overline{x} \in \mathbb{R}^e$ be the average embedding of the untruncated input source text[2].

**NF Latent Module.** To model the variational posterior $q(z \mid x, y; \psi)$, we follow Zhou and Neubig (2017) and assume all the information in $y$ is contained in $x$[3]. Therefore, we have $q(z \mid x, y; \psi) = q(z \mid x; \psi)$, which allows us to parameterize $q(z \mid x; \psi)$ with neural networks (NNs) and normalizing flows using the amortization and reparameterization tricks (Kingma and Welling, 2014). The NF latent module comprises of an inference network $q_0(\cdot)$ and a normalizing flows model. The inference network takes $\overline{x}$ as input and produces two output vectors, $\mu_0 \in \mathbb{R}^\ell$ and $\log(\sigma_0) \in \mathbb{R}^\ell$. Using the reparameterization trick, a random sample $z_0 \in \mathbb{R}^\ell$ is drawn from $N(\mu_0, \text{diag}(\sigma_0^2))$. Afterward, the normalizing flows model applies a sequence of $K$ invertible transformations to $z_0$ to obtain the latent

code $z = z_K = f_K \circ \cdots \circ f_1(z_0) \in \mathbb{R}^\ell$.[4] Note that when $K = 0$, the model reverts to the traditional VED framework, and we refer to this degenerated version as VEDSUM.

**Gated Transformer-based Encoder-Decoder.** Our model adopts the Transformer-based encoder-decoder. The encoder processes the input text and learns a sequence of hidden representations, and the decoder generates a summary based on the encoder's hidden states and the previously generated tokens. We incorporate the latent information into the decoder with a gate mechanism, which mixes the latent vector $z_K$ with the decoder's last layer of hidden states $\{h_j\}_{j=1}^n$. As pointed out in Gu et al. (2020), the saturation property of traditional gating mechanisms hinders gradient-based optimization. Therefore, following their proposal, we use a refined gate mechanism designed to allow for better gradient flow. Let $\sigma(\cdot)$ be the sigmoid function. We generate the gated fused hidden states $\{h_j'\}_{j=1}^n$ as in Eq. 4.

$$z_K' = W^z z_K \in \mathbb{R}^d, \text{ where } W^z \in \mathbb{R}^{d \times \ell}$$
$$f_j = \delta\left(W^f\left[h_j; z_K'\right]\right) \in \mathbb{R}^d, \text{ where } W^f \in \mathbb{R}^{d \times 2d}$$
$$r_j = \delta\left(W^r\left[h_j; z_K'\right]\right) \in \mathbb{R}^d, \text{ where } W^r \in \mathbb{R}^{d \times 2d}$$
$$g_j = (1 - r_j) \cdot f_j^2 + r_j\left(1 - (1 - f_j)^2\right) \in \mathbb{R}^d$$
$$h_j' = (1 - g_j) \cdot h_j + g_j \cdot z_K' \in \mathbb{R}^d$$
$$(4)$$

Afterward, the fused hidden states are passed to a language model (LM) Head layer, where they are transformed into vectors modeling the probabilities of each word in the vocabulary.

### 3.2 Training Objective

Traditional VEDs usually assume $q(z \mid x; \psi)$ to be a Gaussian, allowing analytical computation of the KL term in ELBO. However, in our normalizing flows-based VED, the variational posterior $q(z \mid x) = q_K(z_K \mid x)$ can be complex and hence the KL term in Eq. 3 lacks an analytical form. Therefore, we rewrite the ELBO via a change of variables to enable analytical evaluation[5]:

$$\text{ELBO}_{\text{NF-VED}}$$
$$= \mathbb{E}_{q_0(z_0)}\left[\log p\left(y \mid x, z_K\right) + \log p\left(z_K \mid x\right)\right]$$
$$- \mathbb{E}_{q_0(z_0)}\left[\log q_0\left(z_0\right) - \sum_{k=1}^K \log\left|\det J_{f_k}\left(z_{k-1}\right)\right|\right],$$
$$(5)$$

---

[2]Let $V$ be the vocabulary size, $\{E_v\}_{v=1}^V$ be the input embeddings, and $\{b_v\}_{v=1}^V$ be the Bag-of-Words (BoW) of the input source text, then $\overline{x} = (\sum_{v=1}^V b_v E_v)/(\sum_{v=1}^V b_v) \in \mathbb{R}^e$. In addition, when we don't truncate the input text, $b^T \mathbb{1} = m$ holds. However, if we truncate the input due to encoder constraints, then $b^T \mathbb{1} > m$, and the BoW vector will contain information that would otherwise have been lost.

[3]See detailed discussion in Appendix C.

[4]The log-determinant of the Jacobian at each layer is recorded along the forward call for loss computation.

[5]See derivation in Appendix B Eq. 9.

where $q_0$ is $z_0$'s probability density function, a Gaussian distribution modeled by NNs, and $\det J_{f_k}(\cdot)$ is the determinant of $f_k$'s Jacobian.

Let $\mathcal{L}_{\text{CE}}$ denote the cross-entropy loss and $\mathcal{L}_{\text{VI}}$ denote the loss introduced by the variational latent module. Applying the idea of Monte Carlo to Eq. 5, we obtain the training objective as below. Note that $\mathcal{L}_{\text{VI}}$ is a Monte Carlo estimate of the KL divergence between the variational posterior $q_K$ and the conditional prior distribution $p(z_K \mid x)$.

$$
\begin{aligned}
\mathcal{L} &= \mathcal{L}_{\text{CE}} + \mathcal{L}_{\text{VI}} \\
&= -\sum_{j=1}^{n} \log p\left(y_j \mid \{x_i\}_{i=1}^{m}, z_K, y_{<j}\right) \\
&\quad + \log q_0\left(z_0\right) - \sum_{k=1}^{K} \log |\det J_{f_k}\left(z_{k-1}\right)| \\
&\quad - \log p\left(z_K \mid x\right)
\end{aligned}
\tag{6}
$$

### 3.3 Mitigating Posterior Collapse

To remedy posterior collapse, we consider two strategies, aiming to preserve the expressiveness of the latent variable and improve the overall summary quality. The first approach, called $\beta_C$-VAE (Prokhorov et al., 2019), replaces the KL term with $\beta |KL - C|$, where $\beta$ is a scaling factor, and $C \geq 0$ is a threshold that regulates the magnitude of the KL term. When $C > 0$, the KL term is expected to be discouraged from getting close to 0.

We propose the second approach, Controlled Alternate Aggressive Training (CAAT), inspired by the lagging inference strategy (He et al., 2019). This strategy uses the observation that the inference network cannot accurately approximate the true posterior in the initial stages of training. As outlined in Alg. 1 in Appendix A, CAAT comprises two stages. In the first stage, we alternately update the variational parameters and the entire parameters[6] for a specified number of steps. In the second stage, we train all parameters jointly, as in basic VAE training, for the remainder of the training.

### 3.4 NF-enhanced Knowledge Distillation

Normalizing flows can learn complex and multimodal distributions (Papamakarios et al., 2017), which makes them a promising approach for knowledge distillation tasks that involve integrating information from multiple sources (Hinton et al., 2015). To investigate the impact of normalizing flows on knowledge distillation, we adopt two knowledge distillation methods by Shleifer and Rush

(2020): Shrink and Fine-Tune (SFT) and Pseudo-labels (PL). SFT shrinks the teacher model and re-finetunes the shrunk model. In contrast, the PL method initializes the student model with the compressed version produced by SFT and then fine-tunes using the pseudo-labeled data generated by the teacher model. In this study, we fine-tune the model on the augmented data with both original and pseudo-labeled data, enabling it to more effectively switch between generated summaries and ground truth, thereby mitigating exposure bias.

## 4 Experiments

### 4.1 Datasets

We evaluate the effectiveness of FlowSUM on six public benchmark datasets[7], including CNN/Daily Mail (CNN/DM) (Hermann et al., 2015), XSum (Narayan et al., 2018), Multi-News (Fabbri et al., 2019), arXiv, PubMed (Cohan et al., 2018), and SAMSum (Gliwa et al., 2019). These datasets exhibit various summary styles and lengths, and their corresponding statistics are shown in Table 1. Refer to Appendix E for more details.

| Datasets | Split (train/val/test) | Avg. doc length | Avg. summary length |
|---|---|---|---|
| CNN/DM | 287113/13368/11490 | 781 | 56 |
| Multi-News | 44972/5622/5622 | 2103 | 264 |
| arXiv | 203037/6436/6440 | 4938 | 220 |
| PubMed | 119924/6633/6658 | 3016 | 203 |
| XSum | 204045/11332/11334 | 431 | 23 |
| SAMSum | 14732/818/819 | 94 | 20 |

Table 1: Statistics of Summarization Datasets.

### 4.2 Implementation Details

We configure the inference net $q_0(z_0|\bar{x})$ to be a feedforward neural network and set the latent dimension $\ell$ to 300 and the number of NF layers $K \in \{2, 4, 6, 8\}$. For models that use $\beta_C$-VAE, we set $\beta = 1$ and $C = 0.1$, and for those using CAAT, we conduct one epoch of aggressive training with $n_{alt} = 15$ and two epochs of non-aggressive training. See more details in Appendix G.

### 4.3 Baselines

We use BART (Lewis et al., 2020) and BERT2BERT (Rothe et al., 2020) as two backbone models. We refer to the PL knowledge distilled

---

[6]In our preliminary experiments, we find that if we alternate between variational and encoder-decoder parameters, the training becomes unstable and generates NaN values. Therefore, we alternate between variational and all parameters.

[7]We access them through Hugging Face Datasets, which provides reproducible code for processing texts and generating train/validation/test splits.

FlowSUM as FlowSUM-PLKD. Our comparison involves the following baselines: PG+Cov (See et al., 2017), BERT2BERT (Rothe et al., 2020), BERTSUM (Liu and Lapata, 2019), BART (Lewis et al., 2020), PEGASUS (Zhang et al., 2020a), VHTM (Fu et al., 2020), TAS (Zheng et al., 2020), and PEGASUS+Flow-NTM (Nguyen et al., 2021). See Appendix F for more detailed descriptions.

### 4.4 Results

#### 4.4.1 Automatic Evaluation

We evaluate the generated summary quality using ROUGE scores (Lin, 2004) and BERTScore (Zhang et al., 2020b)[8]. Specifically, we utilize the overlap of unigrams and bigrams (ROUGE-1 and ROUGE-2) to evaluate the informativeness, and the longest common subsequence (ROUGE-L) for fluency. Moreover, we report BERTScore, which gauges semantic similarity based on contextual embeddings. Furthermore, we present rep-w (Fu et al., 2021)[9] and the average length of summaries to gain a better understanding of the quality.

We compare the proposed model against baseline models in ROUGE scores in Tables 2 and 3. On CNN/DM, FlowSUM (BERT2BERT) greatly outperforms BERT2BERT, whereas VEDSUM adds noise to the model and leads to a decrease in performance. With the BART backbone, FlowSUM achieves an absolute improvement over the BART model with +0.48, +0.08, and +0.75 in R-1, 2, and L scores, respectively. However, on XSum, the variational models do not perform well when the gold summaries involve only one sentence. VEDSUM leads to a significant decrease in performance, whereas with FlowSUM, the decrease in ROUGE scores is less severe, leading to +0.12, -0.15, and -0.25 in R-1, 2, and L scores, respectively.

Table 4 uses BART as the backbone and compares BART, VEDSUM, and FlowSUM across all datasets. Overall, variational models produce summaries of superior quality for datasets with long summaries, such as CNN/DM, Multi-News, arXiv, and PubMed, and FlowSUM further enhances the performance beyond VEDSUM. However, when it comes to datasets featuring short summaries such as XSum and SAMSum, the variational component markedly diminishes the model performance.

---

[8]We obtain both metrics using Hugging Face Evaluate and report the $F_1$ scores.

[9]rep-w is calculated as the proportion of the current token that appears in the previous $w$ tokens. Refer to Appendix D for the detailed definition.

| Model | ROUGE ↑ | | |
|---|---|---|---|
| | 1 | 2 | L |
| PG+Cov (See et al., 2017) | 39.53 | 17.28 | 36.38 |
| BERT2BERT (Rothe et al., 2020) | 41.28 | 18.69 | 38.09 |
| BERTSUM (Liu and Lapata, 2019) | 42.13 | 19.60 | 39.18 |
| BART (Lewis et al., 2020) | 44.16 | 21.28 | 40.90 |
| PEGASUS (Zhang et al., 2020a) | 44.17 | 21.47 | 41.11 |
| VHTM (Fu et al., 2020) | 40.57 | 18.05 | 37.18 |
| TAS (Zheng et al., 2020) | 44.38 | 21.19 | 41.33 |
| PEGASUS+NTM (Nguyen et al., 2021) | 44.52 | **21.95** | 41.39 |
| VEDSUM (BERT2BERT) | 40.89 | 18.28 | 37.95 |
| FlowSUM (BERT2BERT) | 41.51 | 18.81 | 38.56 |
| VEDSUM (BART) | 44.36 | 21.09 | 41.37 |
| FlowSUM (BART) | **44.64** | 21.36 | **41.65** |
| FlowSUM-PLKD (BART) | 44.59 | 21.49 | 41.59 |

Table 2: Comparison with baselines on CNN/DM.

| Model | ROUGE ↑ | | |
|---|---|---|---|
| | 1 | 2 | L |
| PG+Cov (See et al., 2017) | 28.10 | 8.02 | 21.72 |
| BERTSUM (Liu and Lapata, 2019) | 38.81 | 16.50 | 31.27 |
| BART (Lewis et al., 2020) | 45.14 | 22.27 | 37.25 |
| PEGASUS (Zhang et al., 2020a) | 47.21 | 24.56 | 39.25 |
| TAS (Zheng et al., 2020) | 44.63 | 21.62 | 36.77 |
| PEGASUS+NTM (Nguyen et al., 2021) | **49.57** | **25.08** | **41.81** |
| VEDSUM (BART) | 43.62 | 20.27 | 35.06 |
| FlowSUM (BART) | 45.26 | 22.12 | 37.00 |
| FlowSUM-PLKD (BART) | 45.54 | 22.67 | 37.38 |

Table 3: Comparison with baselines on XSum.

We hypothesize that brief summaries may be more susceptible to disturbances and are more prone to being affected by noise. Nevertheless, incorporating NF modules alleviates these reductions and accomplishes comparable outcomes. Furthermore, we observe that both variational models tend to generate lengthier summaries, while FlowSUM exhibits fewer issues with repetition compared to VEDSUM.

#### 4.4.2 On NF-enhanced Knowledge Distillation

We use PEGASUS as the teacher model to generate pseudo-labels on the CNN/DM training set. In this study, we explore the effects of knowledge distillation on BART and DistilBART, a shrunken version of BART. We examine two variations of Distil-BART: dBART-6-6, which replicates 6 layers[10] of the BART encoder and decoder, and dBART-12-3, which duplicates all layers of the BART encoder and 3 layers[11] of the decoder.

Table 5 presents the impact of the PL approach on the original BART model. Training the BART

---

[10]The 0, 2, 4, 7, 9, and 11th layer.

[11]The 0, 6, and 11th layer.

| Model | ROUGE ↑ 1/2/L | BERT-Score ↑ | rep-w ↓ | Length |
|---|---|---|---|---|
| **CNN/DM** | | | | |
| BART | 44.16/21.28/40.90 | 89.40 | **8.31** | 84.11 |
| VEDSUM | 44.34/21.09/41.37 | 89.20 | 8.43 | 88.63 |
| FlowSUM | **44.64/21.36/41.65** | **89.46** | 8.43 | 92.24 |
| **Multi-News** | | | | |
| BART | 42.56/15.34/36.67 | 86.69 | **9.76** | 133.42 |
| VEDSUM | 43.91/16.68/38.10 | 87.04 | 9.95 | 128.79 |
| FlowSUM | **44.42/17.01/38.36** | **87.09** | 9.91 | 128.87 |
| **arXiv** | | | | |
| BART | 42.55/15.92/37.89 | 85.35 | 17.23 | 130.68 |
| VEDSUM | 43.05/**16.34**/38.26 | 85.44 | 16.63 | 130.92 |
| FlowSUM | **43.11**/16.26/**38.31** | **85.45** | **16.55** | 132.88 |
| **PubMed** | | | | |
| BART | 41.57/16.72/36.94 | 84.65 | 13.26 | 136.10 |
| VEDSUM | 44.21/19.20/39.32 | 85.07 | 12.76 | 138.70 |
| FlowSUM | **44.55/19.50/39.59** | **85.16** | **12.59** | 138.09 |
| **XSum** | | | | |
| BART | 45.14/**22.27/37.25** | **92.16** | **4.63** | 25.54 |
| VEDSUM | 43.62/20.27/35.06 | 91.75 | 5.96 | 31.22 |
| FlowSUM | **45.26**/22.12/37.00 | 92.13 | 4.95 | 28.71 |
| **SAMSum** | | | | |
| BART | **53.16**/28.19/**49.03** | **92.68** | 6.71 | 30.00 |
| VEDSUM | 51.91/26.74/47.41 | 92.40 | 7.53 | 30.92 |
| FlowSUM | 53.13/**28.49**/49.00 | 92.67 | **6.59** | 29.77 |

Table 4: Comparison of BART, VEDSUM (BART), and FlowSUM (BART) on all six benchmarks.

| Model | ROUGE ↑ | | | BERT-Score ↑ | Length |
|---|---|---|---|---|---|
| | 1 | 2 | L | | |
| BART | 44.16 | 21.28 | 40.90 | 89.40 | 84.11 |
| VEDSUM | 44.34 | 21.09 | 41.37 | 89.20 | 88.63 |
| FlowSUM (Planar) | 44.62 | 21.32 | 41.64 | 89.20 | 90.78 |
| FlowSUM (RQNSF) | **44.64** | 21.36 | **41.65** | 89.46 | 92.24 |
| PEGASUS | 44.17 | 21.47 | 41.11 | **89.52** | 77.84 |
| BART-PLKD | 42.83 | 20.16 | 39.98 | 89.04 | 100.52 |
| VEDSUM-PLKD | 44.45 | 21.25 | 41.45 | 89.41 | 93.42 |
| FlowSUM-PLKD (Planar) | 44.19 | 21.03 | 41.15 | 89.34 | 92.38 |
| FlowSUM-PLKD (RQNSF) | 44.59 | **21.48** | 41.59 | 89.47 | 84.75 |

Table 5: PL Knowledge Distillation on BART on CNN/DM.

| Model | ROUGE ↑ 1/2/L | BERT-Score ↑ | Length | # Params (MM) | Inference Time (MS) ↓ |
|---|---|---|---|---|---|
| **dBART-6-6** | | | | | |
| dBART-6-6 | 42.78/20.24/39.72 | 88.98 | 67.42 | 230 | 170.5 |
| FlowSUM | 43.41/20.33/40.41 | 89.18 | 91.25 | 238 | 234.9 |
| FlowSUM-PLKD | **43.70/20.71/40.73** | **89.24** | 91.10 | 238 | 239.7 |
| **dBART-12-3** | | | | | |
| dBART-12-3 | 43.39/20.57/40.44 | 89.20 | 85.48 | 255 | 199.6 |
| FlowSUM | 43.53/20.61/40.59 | 89.28 | 83.74 | 263 | 190.7 |
| FlowSUM-PLKD | **44.05/21.06/41.07** | **89.37** | 84.48 | 263 | 200.4 |

Table 6: Knowledge Distillation on DistilBART on CNN/DM.

model on augmented data worsens the performance compared to training on the original data. In contrast, VEDSUM-PLKD achieves improvements in all three ROUGE scores, and FlowSUM-PLKD with RQNSF achieves the highest R-2 score, albeit with some sacrifice in R-1 and R-L[12]. However, planar flows appear to be unsuitable for knowledge distillation via PL. To better understand FlowSUM-PLKD, we visualize the latent distribution (see Appendix I) and demonstrate how the NF's ability to capture multi-modality could account for its impressive performance.

Table 6 investigates the two DistilBART variants with RQNSF. With FlowSUM, both variants achieve improvements, suggesting that NF is beneficial for the SFT approach. Previous experiments from Shleifer and Rush (2020) showed that PL performed worse than SFT on CNN/DM. However, our experiments reveal that the NF latent module unleashes the potential of PL. When trained on augmented data, FlowSUM-PLKD (dBART-6-6)

achieves R-1/2/L improvements of 0.92/0.47/1.01 over dBART-6-6, and FlowSUM-PLKD (dBART-12-3) achieves improvements of 0.66/0.49/0.63 over dBART-12-3, much more than the SFT approach. Furthermore, FlowSUM does not introduce additional computational burden at inference, and the time cost is primarily related to the length of the generated summaries.

### 4.4.3 Analysis on NF Types and Depth

We investigate the effect of NF types and the number of NF layers on the Multi-News dataset[13]. Table 7 explores the effect of NF types. Simple flows like Planar and Radial yield inferior performance compared to the VAE counterpart, whereas more complex flows tend to achieve greater improvements. Overall, IAF and RQNSF emerge as the best-performing NF types.

Table 8 delves further into IAF and RQNSF, investigating the effect of NF depth. The findings indicate that adding more layers does not always lead to improved performance. We hypothesize that when the encoder-decoder model is well-trained, the increased complexity of the NF module may introduce more noise, outweighing the benefits of better latent modeling and subsequently worsening the summary quality.

---

[12]This can be explained by the teacher model's worse performance in these two metrics.

[13]We choose Multi-News due to its smaller size, enabling us to conduct experiments with reduced computational cost.

| Model | ROUGE ↑ 1/2/L | BERT-Score ↑ | rep-w ↓ | Length |
|---|---|---|---|---|
| BART | 42.56/15.35/36.67 | 86.69 | **9.76** | 133.42 |
| VEDSUM | 43.91/16.68/38.10 | 87.04 | 9.95 | 128.79 |
| FlowSUM (Planar) | 43.85/16.61/37.97 | 87.03 | 10.04 | 128.84 |
| FlowSUM (Radial) | 43.84/16.68/37.98 | 87.04 | 9.92 | 128.72 |
| FlowSUM (Sylvester) | 44.18/16.71/38.15 | 87.08 | 9.80 | 128.76 |
| FlowSUM (RealNVP) | 44.19/16.64/38.15 | 87.05 | 9.81 | 128.76 |
| FlowSUM (IAF) | **44.42/17.01/38.36** | **87.09** | 9.91 | 128.87 |
| FlowSUM (RLNSF) | 44.25/16.86/38.14 | 87.06 | 9.80 | 128.80 |
| FlowSUM (RQNSF) | 44.31/16.98/38.27 | 87.07 | 9.91 | 128.81 |

Table 7: Effect of NF Types on Multi-News.

| Model | ROUGE ↑ 1/2/L | BERT-Score ↑ | rep-w ↓ | Length |
|---|---|---|---|---|
| FlowSUM (IAF-4) | 44.30/**17.03**/38.22 | 87.05 | **9.82** | 128.81 |
| FlowSUM (IAF-6) | **44.42**/17.01/**38.36** | **87.09** | 9.91 | 128.87 |
| FlowSUM (IAF-8) | 44.18/16.90/38.16 | 87.04 | 9.88 | 128.84 |
| FlowSUM (RQNSF-2) | 44.15/16.88/38.20 | 87.04 | 9.94 | 128.83 |
| FlowSUM (RQNSF-4) | **44.31/16.98/38.27** | **87.07** | 9.91 | 128.81 |
| FlowSUM (RQNSF-6) | 44.15/16.88/38.18 | 87.06 | **9.87** | 128.92 |

Table 8: Effect of Number of NF Layers on Multi-News.

| Model | Training | ROUGE ↑ | | | KL Divergence |
|---|---|---|---|---|---|
| | | 1 | 2 | L | |
| VEDSUM | standard | 43.91 | 16.68 | 38.10 | 0.0117 |
| VEDSUM | $\beta_C$-VAE | 43.78 | 16.54 | 37.96 | 0.0082 |
| FlowSUM (Planar) | standard | 43.85 | 16.61 | 37.97 | 0.2719 |
| FlowSUM (Planar) | $\beta_C$-VAE | 43.68 | 16.47 | 37.85 | 0.1815 |
| FlowSUM (Radial) | standard | 43.63 | 16.37 | 37.82 | 0.0121 |
| FlowSUM (Radial) | $\beta_C$-VAE | 43.84 | 16.68 | 37.98 | 0.0096 |
| FlowSUM (Sylvester) | standard | 43.68 | 16.51 | 37.87 | 0.0841 |
| FlowSUM (Sylvester) | $\beta_C$-VAE | 44.18 | 16.71 | 38.15 | 0.0348 |
| FlowSUM (RealNVP) | standard | 44.19 | 16.64 | 38.15 | 4.7986 |
| FlowSUM (RealNVP) | $\beta_C$-VAE | 43.71 | 16.54 | 37.85 | 7.8938 |
| FlowSUM (RealNVP) | CAAT | 44.12 | 16.82 | 38.11 | 5.2107 |
| FlowSUM (IAF) | standard | 43.87 | 16.62 | 37.97 | 3.9146 |
| FlowSUM (IAF) | $\beta_C$-VAE | 43.81 | 16.58 | 37.91 | 3.9128 |
| FlowSUM (IAF) | CAAT | 44.30 | 17.03 | 38.22 | 2.1108 |
| FlowSUM (RLNSF) | standard | 44.25 | 16.86 | 38.14 | 104.9667 |
| FlowSUM (RLNSF) | $\beta_C$-VAE | 44.25 | 16.86 | 38.14 | 104.9667 |
| FlowSUM (RLNSF) | CAAT | 44.14 | 16.82 | 38.05 | 95.3774 |
| FlowSUM (RQNSF) | standard | 44.18 | 16.76 | 38.18 | 127.8106 |
| FlowSUM (RQNSF) | $\beta_C$-VAE | 44.18 | 16.76 | 38.18 | 127.8106 |
| FlowSUM (RQNSF) | CAAT | 44.31 | 16.98 | 38.27 | 107.0794 |

[a] VEDSUM and FlowSUM with radial flows have no CAAT results as the training is unstable and generates NaN values.

Table 9: Effect of Training Strategies.

| Training | Gate Initialization | ROUGE ↑ | | |
|---|---|---|---|---|
| | | 1 | 2 | L |
| standard | standard | 40.82 | 18.29 | 37.92 |
| standard | near-zero | 40.98 | 18.36 | 38.09 |
| CAAT | standard | **41.51** | **18.81** | **38.56** |
| CAAT | near-zero | 41.13 | 18.57 | 38.21 |

Table 10: Effect of CAAT and Gate Initialization.

### 4.4.4 Analysis on Training Strategies

We implement standard VAE training, $\beta_C$-VAE, and CAAT on VEDSUM and FlowSUM models, and we evaluate their effectiveness with different types of normalizing flows. Table 9 shows that VEDSUM and FlowSUM models with residual flows, including planar, radial, and Sylvester flows, suffer from posterior collapse, whereas those with more complex flows do not. Moreover, applying $\beta_C$-VAE to VEDSUM and FlowSUM models with residual flows does not effectively mitigate posterior collapse but even exacerbates the issue. Furthermore, for models with planar, RealNVP, and IAF flows, training with $\beta_C$-VAE worsens ROUGE scores, while for radial and Sylvester flows, it improves performance. Notably, the two neural spline flows are not impacted by $\beta_C$-VAE training.

Concerning CAAT, we note that applying it to treat severe posterior collapses such as VEDSUM and FlowSUM with residual flows can cause instability in training while producing NaN values. Hence, it is only effective for models with KL divergence that is not close to zero. Nonetheless, when applicable, CAAT enhances the quality of summaries, particularly when utilized with the top-performing NFs, namely IAF and RQNSF.

In addition, we explore the impact of gate score initialization. The standard method initializes gating weights with small deviations from zero, resulting in an initial gate score close to 0.5. In contrast, the near-zero initialization method initializes gating weights such that the resulting gate score is approx-imately 0.05. Our experiments using FlowSUM (BERT2BERT) with RQNSF as the base model reveal that CAAT + Standard Gate Score Initialization yields the best results and the most stable training process, as illustrated in Table 10 and Figures 2 to 3 in Appendix H. This suggests that by setting a large initial gate score and forcing the model to learn from the NF latent module, we can better capture latent code information.

## 5 Conclusions and Discussions

This paper introduces FlowSUM, a normalizing flows-based Variational Encoder-Decoder (VED) framework for text summarization. It outperforms a leading non-latent model across multiple datasets. This enhanced performance is attributed to the flexible posterior distributions provided by normalizing flows. We also analyze the operating characteristics and the posterior collapse problem of normalizing flows and propose an effective training strategy for complex flows. Moreover, we demonstrate that in-

corporating normalizing flows is highly effective for knowledge distillation with minimal impact on inference time.

FlowSUM illustrates the advantages of incorporating flexible latent modeling. Considering the remarkable achievements of Latent Diffusion Models (LDMs) in generating images (Rombach et al., 2022), adopting LDMs for capturing latent representation may produce comparable or even superior outcomes in text summarization. In this scenario, the gating mechanism may not be an appropriate choice. A direct correlation between the latent vector and the target text may be more suitable for executing the diffusion process. Enhancing the architecture to leverage diffusion models could be a potential avenue for future research.

## Limitations

FlowSUM has demonstrated excellent results on datasets with long summaries. However, its performance on short-summary datasets like XSum and SAMSum has been unsatisfactory. The underlying cause could be attributed to suboptimal hyperparameter tuning or the incompatibility of FlowSUM with short summaries. Additional investigations are needed to identify the root cause.

Furthermore, we did not fine-tune the hyperparameters of the normalizing flows model, such as the latent dimension, the number of bins in spline coupling layers, and the neural network in IAF, RealNVP, RLNSF, and RQNSF. Moreover, we opted for a small batch size due to memory limitations. Adjusting these hyperparameters could potentially enhance the model's performance.

Due to limited computational resources, we utilized BART and BERT2BERT as the backbone models instead of newer architectures. Further research may focus on verifying the effectiveness of FlowSUM on more advanced structures.

## Ethics Statement

Our research entailed developing a new text summarization framework. Although no private data were utilized, we acknowledge the potential societal impacts of our work. Therefore, we adhered to pertinent ethical guidelines and implemented rigorous procedures to guarantee the accuracy of our results.

## Acknowledgements

This work was supported in part by NSF grant DMS-1952539 and NIH grants R01AG069895, R01AG065636, R01AG074858, U01AG073079.

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

## A  Controlled Alternate Aggressive Training (CAAT)

---

**Algorithm 1** Controlled Alternate Aggressive Training (CAAT)

---

**Input:** number of aggressive training steps $n_{agg}$; maximum number of training steps $n_{max}$; number of alternating steps $n_{alt}$.

1: $\boldsymbol{\theta}, \boldsymbol{\psi} \leftarrow$ Initialize encoder-decoder parameters and variational parameters respectively
2: **for** $i = 1, 2, \cdots, n_{agg}$ **do**
3:     $\mathbf{X} \leftarrow$ Random data minibatch
4:     **if** $i \mod n_{alt} = 0$ **then**
5:         Compute $\boldsymbol{g}_{\boldsymbol{\theta},\boldsymbol{\psi}} \leftarrow \nabla_{\boldsymbol{\psi},\boldsymbol{\theta}} \mathcal{L}(\mathbf{X}; \boldsymbol{\theta}, \boldsymbol{\psi})$
6:         Update $\boldsymbol{\theta}, \boldsymbol{\psi}$ using gradients $\boldsymbol{g}_{\boldsymbol{\theta},\boldsymbol{\psi}}$
7:     **else**
8:         Compute $\boldsymbol{g}_{\boldsymbol{\psi}} \leftarrow \nabla_{\boldsymbol{\psi}} \mathcal{L}(\mathbf{X}; \boldsymbol{\theta}, \boldsymbol{\psi})$
9:         Update $\boldsymbol{\psi}$ using graidents $\boldsymbol{g}_{\boldsymbol{\psi}}$
10: **for** $i = n_{agg}, n_{agg} + 1, \cdots, n_{max}$ **do**
11:     $\mathbf{X} \leftarrow$ Random data minibatch
12:     Compute $\boldsymbol{g}_{\boldsymbol{\theta},\boldsymbol{\psi}} \leftarrow \nabla_{\boldsymbol{\psi},\boldsymbol{\theta}} \mathcal{L}(\mathbf{X}; \boldsymbol{\theta}, \boldsymbol{\psi})$
13:     Update $\boldsymbol{\theta}, \boldsymbol{\psi}$ using gradients $\boldsymbol{g}_{\boldsymbol{\theta},\boldsymbol{\psi}}$
14:     **if** early stopping criterion is met **then**
15:         **break**

---

Another advantage of the controlled alternate aggressive training (CAAT) strategy is that it provides us with more control. It is commonly assumed that

allowing the model more freedom to learn, even if the NF latent module is not helpful, will not harm performance. However, our experiments suggest that this assumption does not hold, particularly for short-summary datasets where the model will not learn on its own to avoid hurting the original performance. The CAAT strategy allows us to effectively freeze the encoder-decoder parameters by setting $n_{agg}$ and $n_{alt}$ to large values, ensuring that when the nf module is unhelpful, it will not significantly harm performance.

## B  Deeper Dive into the Evidence Lower Bound (ELBO)

Within the VED framework, the conditional data generation process can be expressed as follows:

$$p(y \mid x; \phi, \theta) = \int p(z \mid x; \phi) p(y \mid x, z; \theta) dz.$$

The subsequent challenge revolves around parameter estimation. Typically, the conditional latent prior is assumed as $p(z \mid x; \phi) = N(0, I)$ for simplification (hence eliminating the $\phi$ parameter). Despite this, the likelihood $p(y \mid x; \theta)$ remains computationally intractable to evaluate. Variational inference tackles this issue by introducing a variational distribution $q(z \mid x, y; \psi)$ from a specific parametric family, aiming to approximate the actual posterior $p(z \mid x, y)$. Here, $\theta$ denotes the model parameters, and $\psi$ refers to the variational parameters. Instead of attempting to estimate $\theta$ solely through maximizing the challenging log-likelihood, the approach involves joint estimation of both $\theta$ and $\psi$ by optimizing the ELBO.

Examining Eq. 7 and 8, it's evident that the ELBO represents a lower bound of the log-likelihood. Moreover, a smaller value of $\mathrm{KL}(q(z \mid x, y) \| p(z \mid x, y))$ indicates a closer alignment between the variational posterior and the true posterior, thereby bringing the ELBO closer to the log-likelihood. This insight propels the adoption of normalizing flows to model a flexible family of variational posterior.

$$\begin{aligned}
&\mathrm{KL}(q(z \mid x, y) \| p(z \mid x, y)) \\
=& \mathbb{E}_{q(z\mid x,y)}[\log q(z \mid x, y)] - \mathbb{E}_{q(z\mid x,y)}\left[\log \frac{p(z, x, y)}{p(x, y)}\right] \\
=& \mathbb{E}_{q(z\mid x,y)}[\log q(z \mid x, y)] \\
&\quad - \mathbb{E}_{q(z\mid x,y)}\left[\log \frac{p(z, x, y)}{p(x, z)} \cdot \frac{p(x, z)}{p(x)} \cdot \frac{p(x)}{p(x, y)}\right] \\
=& \mathbb{E}_{q(z\mid x,y)}[\log q(z \mid x, y)] - \mathbb{E}_{q(z\mid x,y)}[\log p(y \mid x, z)] \\
&\quad - \mathbb{E}_{q(z\mid x,y)}[\log p(z \mid x)] + \mathbb{E}_{q(z\mid x,y)}[\log p(y \mid x)] \\
=& KL(q(z \mid x, y) \| p(z \mid x)) - \mathbb{E}_{q(z\mid x,y)}[\log p(y \mid x, z)] \\
&\quad + \mathbb{E}_{q(z\mid x,y)}[\log p(y \mid x)] \\
\geqslant& 0
\end{aligned} \tag{7}$$

$$\begin{aligned}
&\mathrm{ELBO_{VED}} \\
=& \mathbb{E}_{q(z\mid x,y)}[\log p(y \mid x, z)] - KL(q(z \mid x, y) \| p(z \mid x)) \\
=& \log p(y \mid x) - \mathrm{KL}(q(z \mid x, y) \| p(z \mid x, y)) \\
\leq& \log p(y \mid x)
\end{aligned} \tag{8}$$

$$\begin{aligned}
&\mathrm{ELBO_{NF\text{-}VED}} \\
=& \mathbb{E}_{q(z\mid x)}[\log p(y \mid x, z)] + \mathbb{E}_{q(z\mid x)} \log p(z \mid x) \\
&\quad - \mathbb{E}_{q(z\mid x)}[\log q(z \mid x)] \\
=& \mathbb{E}_{q_0(z_0)}[\log p(y \mid x, z_K) + \log p(z_K \mid x)] \\
&\quad - \mathbb{E}_{q_0(z_0)}[\log q_K(z_K)] \\
=& \mathbb{E}_{q_0(z_0)}[\log p(y \mid x, z_K) + \log p(z_K \mid x)] \\
&\quad - \mathbb{E}_{q_0(z_0)}\left[\log q_0(z_0) - \sum_{k=1}^{K} \log |\det J_{f_k}(z_{k-1})|\right],
\end{aligned} \tag{9}$$

where $q_0$ and $q_K$ are the probability density function for $z_0$ and $z_K$ respectively.

## C  Discussion on $q(z \mid x, y) = q(z \mid x)$

we choose to assume $q(z \mid x, y) = q(z \mid x)$ for the following reasons. Firstly, this assumption is grounded in the nature of summarization, where $y$ can be viewed as a condensed form of $x$ and hence it is sensible to assume all the information in $y$ is contained in $x$. Secondly, as evidenced by Zhang et al. (2016), it is plausible to condition the posterior on both $x$ and $y$. However, their approach suffers from difficulties during prediction. In prediction, the target text $y$ is not accessible, making it hard to sample from $q(z \mid x, y)$. Zhang et al. (2016) suggests taking the prior's mean as the latent code, but in our paper, the prior is a Gaussian whereas the posterior is a complex distribution modeled by normalizing flows, and taking such a strategy would diminish the benefit of using normalizing flows. Thirdly, it has been shown empirically by Eikema and Aziz (2019) that by restricting the conditioning of the posterior to $x$ alone, their model

achieves higher accuracy. Therefore, we consider $q(z \mid x, y) = q(z \mid x)$ as our modeling strategy.

## D  Repetition Measures

Let $s$ represent the sentences in a result set $\mathcal{D}$, $|s|$ be the number of tokens in $s$, $s_t$ be the $t$th token, and $s_{i:j}$ be the sub-sequence of $s$ from the $i$th token to the $j$th token. The rep-w (Fu et al., 2021) is then defined by Equation 10.

$$\text{rep-w} = \frac{1}{|\mathcal{D}|} \sum_{s \in \mathcal{D}} \frac{1}{|s|} \sum_{t=2}^{|s|} \mathbb{1} \left[ s_t \in s_{\max(t-w,1):t-1} \right] \tag{10}$$

## E  Datasets

**CNN/Daily Mail** (Hermann et al., 2015) consists of 312,085 online news articles, with one article paired with a multi-sentence summary. We use the non-anonymized version as in See et al. (2017) and follow the text processing[14] in Lewis et al. (2020).
**XSum** (Narayan et al., 2018) contains 227k BBC articles, each summarized in a single sentence.
**Multi-News** (Fabbri et al., 2019) is a multi-document dataset comprising 56k pairs of news articles and multi-sentence summaries.
**arXiv, PubMed** (Cohan et al., 2018) are two scientific paper document datasets from arXiv.org (113k) and PubMed (215k). Each pair consists of a scientific article's body document and its abstract.
**SAMSum** (Gliwa et al., 2019) includes 16k conversations annotated with summaries by linguists. Unlike structured texts, the information in dialogues is scattered across different speakers' utterances, increasing the summarization difficulty.

## F  Baseline Models

**PG+Cov** (See et al., 2017) is a pointer-generator (PG) network supplemented with a coverage mechanism that addresses the Out-Of-Vocabulary problem and minimizes word repetition.
**BERT2BERT** (Rothe et al., 2020) initializes both the encoder and the decoder with the pre-trained BERT checkpoints and adds cross-attention layers.
**BERTSUM** (Liu and Lapata, 2019) builds on top of BERT and applies a fine-tuning scheduler to better align the encoder and the decoder.
**BART** (Lewis et al., 2020) is a pretrained denoising autoencoder with the standard sequence-to-

---

[14]We update the data loading script following https://github.com/facebookresearch/fairseq/issues/1401.

sequence Transformer architecture. In this paper, we use BART as the encoder-decoder backbone.
**PEGASUS** (Zhang et al., 2020a) is a large Transformer-based S2S model, pre-trained on massive text data using a self-supervised objective called gap sentence generation, designed for abstractive summarization.
**VHTM** (Fu et al., 2020) is a variational hierarchical model built on the PG network. It models the topic proportion vector with isotropic Gaussian and fuses in topic information at diverse granularity levels.
**TAS** (Zheng et al., 2020) is a topic-guided Transformer-based S2S model that injects the topic-word matrix into the LMHead layer and jointly trains the NTM and encoder-decoder model.
**PEGASUS+Flow-NTM** (Nguyen et al., 2021) is a topic-aware model built on PEGASUS. It utilizes a Flow-based NTM and a contextualized gating mechanism to integrate topic information into the encoder and the decoder.

## G  Implementation Details

### G.1  NF Latent Module

We configure the inference net $q(z_0|\overline{x})$ to be a feed-forward neural network with three hidden layers of dimension $\in \{300, 600\}$, Tanh activations, and a 0.1 dropout rate. We set the latent dimension $\ell$ to 300 and the number of NF layers $\in \{2, 4, 6, 8\}$. For spline coupling layers (RLNSF and RQNSF), we set the number of bins to 4, the bound to 3.0, the split dimension to $\ell/2$, and the neural network to have two hidden layers with the dimension $\ell$. For RealNVP, the split dimension is $\ell/2$, and the neural network has one hidden layer with a dimension of $10\ell$. For IAF, the neural network features one hidden layer of the dimension $3\ell + 1$. Moreover, we set $\beta = 1$ and $C = 0.1$ for models that use $\beta_C$-VAE, and for models that use CAAT, we conduct one epoch of aggressive training with $n_{alt} = 15$, followed by two epochs of non-aggressive training.

### G.2  Optimization

We train the models using the Adam optimizer (Kingma and Ba, 2015) with $\beta_1 = 0.9, \beta_2 = 0.999$, and $\epsilon = 10^{-8}$. The initial learning rate is $5 \times 10^{-5}$. We employ a linear learning rate scheduler that increases the learning rate from 0 to the initial learning rate during the warmup stage and decreases it from the initial learning rate to 0 after the warmup stage. We also apply the gradient clipping technique with a maximum gradient norm of 1.0. Fur-

thermore, we terminate the training early when the perplexity fails to improve for eight or sixteen consecutive evaluation calls.

## G.3 Model Hyper Parameters

Table 11 provides the hyper-parameters for the models discussed in Table 4 - 7, for the sake of reproducibility. To ensure fair comparisons, unless otherwise specified, the VEDSUM models typically employ the same set of hyper-parameters as their FlowSUM counterparts, except with standard training and no NF layers applied. Additionally, the models in Table 8 have the same hyper-parameters as those in Table 7, except for the number of NF layers used. Lastly, in Table 9, all FlowSUM models use 4 NF layers and the same set of hyper-parameters as those in Table 7 but vary in their training strategies.

## H Experiments on Training Strategies and Gate Initialization

The training curves for the methods in Table 10 are illustrated in Figure 2. The plot demonstrates that the gate score decreases gradually and remains high during aggressive training when CAAT is combined with standard initialization. This combination compels the model to utilize the latent code information effectively. Moreover, as presented in Figure 2c, even though CAAT combined with standard initialization starts with a high perplexity, it achieves a lower perplexity level than other approaches by the end. By examining the training procedure in detail, Figure 3 further indicates that CAAT contributes to greater training stability than standard training.

## I Visualization of Latent Distribution

To gain a better understanding of how normalizing flows contribute to knowledge distillation, we selected several examples from the CNN/Daily Mail and XSum datasets and visualized the resulting latent distribution generated by the FlowSUM-PLKD model, as shown in Figure 4 and 5. For both cases, the transformed latent code $z_K$ exhibited a highly flexible distribution. Notably, in the CNN/Daily Mail example, the first dimension of the second example demonstrated a clear bi-modal distribution, indicating the model's ability to capture information from multiple sources. Similarly, in the XSum dataset examples, we observed distinct multi-modal patterns.

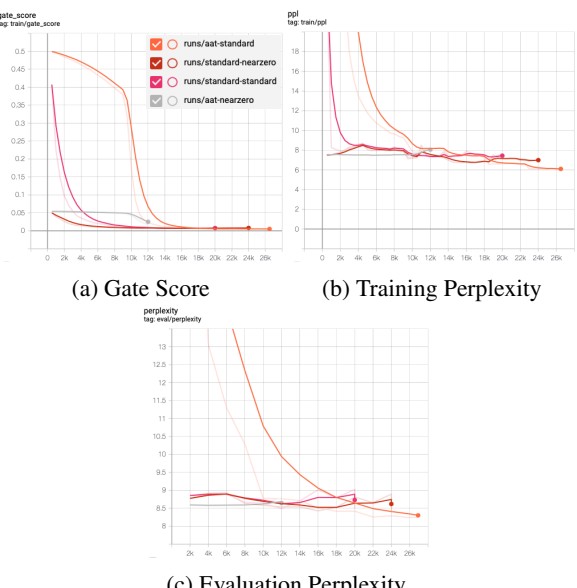

(a) Gate Score      (b) Training Perplexity

(c) Evaluation Perplexity

Figure 2: Comparison of training strategies and gate initialization.

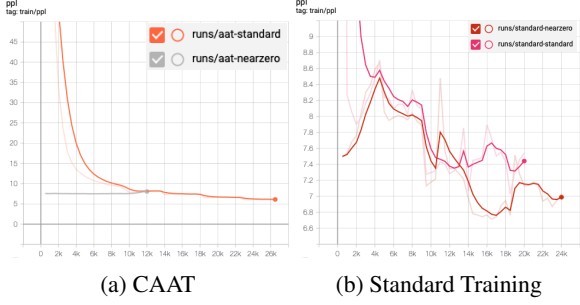

(a) CAAT      (b) Standard Training

Figure 3: A closer look at the training process: CAAT vs. Standard Training.

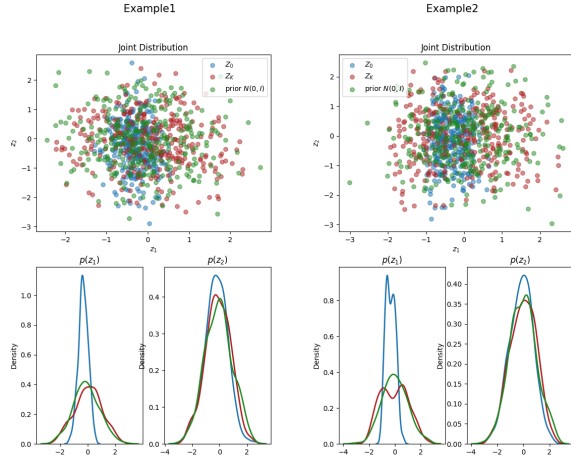

Figure 4: Visualization of the first two dimensions of $z_0$, $z_K$, and $N(0, I)$ by FlowSUM-PLKD on CNN/DM. The right sub-figure demonstrates a clear bi-modality.

| | FlowSUM in Table 4 | | | | | | | | | |
|---|---|---|---|---|---|---|---|---|---|---|
| Dataset | Number of epochs | Number of aggressive epochs | Batch size | Inference net hidden dim | NF type | Number of NF layers | Beam size | Length penalty | Max input tokens | Max target tokens |
| CNN/Daily Mail | 3 | 1 | 8 | 300 | RQNSF | 4 | 4 | 2.0 | 1024 | 128 |
| Multi-News | 3 | 1 | 8 | 600 | IAF | 6 | 4 | 2.0 | 1024 | 128 |
| arXiv | 4 | 1 | 16 | 600 | RQNSF | 4 | 4 | 2.0 | 1024 | 142 |
| PubMed | 4 | 1 | 16 | 600 | RQNSF | 6 | 4 | 2.0 | 1024 | 142 |
| XSum | 3 | 1 | 8 | 600 | RQNSF | 4 | 6 | 0.5 | 1024 | 62 |
| SAMSum | 12 | 12 | 8 | 600 | RQNSF | 4 | 6 | 1.0 | 1024 | 62 |
| | Models in Table 5 | | | | | | | | | |
| Model | Number of epochs | Number of aggressive epochs | Batch size | Inference net hidden dim | NF type | Number of NF layers | Beam size | Length penalty | Max input tokens | Max target tokens |
| VEDSUM | 3 | 0 | 8 | 600 | -[a] | - | 4 | 2.0 | 1024 | 128 |
| FlowSUM (Planar) | 3 | 0 | 8 | 600 | Planar | 4 | 4 | 2.0 | 1024 | 128 |
| FlowSUM (RQNSF) | 3 | 1 | 8 | 300 | RQNSF | 4 | 4 | 2.0 | 1024 | 128 |
| BART-PLKD | 3 | 0 | 8 | - | - | - | 4 | 2.0 | 1024 | 128 |
| VEDSUM-PLKD | 3 | 0 | 8 | 600 | - | - | 4 | 2.0 | 1024 | 128 |
| FlowSUM-PLKD (Planar) | 3 | 0 | 8 | 600 | Planar | 4 | 4 | 2.0 | 1024 | 128 |
| FlowSUM-PLKD (RQNSF) | 3 | 1 | 8 | 300 | RQNSF | 4 | 4 | 2.0 | 1024 | 128 |
| | Models in Table 6 | | | | | | | | | |
| Model | Number of epochs | Number of aggressive epochs | Batch size | Inference net hidden dim | NF type | Number of NF layers | Beam size | Length penalty | Max input tokens | Max target tokens |
| | dBART-6-6 | | | | | | | | | |
| FlowSUM | 3 | 1 | 8 | 300 | RQNSF | 4 | 4 | 2.0 | 1024 | 128 |
| FlowSUM-PLKD | 3 | 1 | 8 | 300 | RQNSF | 4 | 4 | 2.0 | 1024 | 128 |
| | dBART-12-3 | | | | | | | | | |
| FlowSUM | 3 | 1 | 8 | 300 | RQNSF | 4 | 4 | 2.0 | 1024 | 128 |
| FlowSUM-PLKD | 3 | 1 | 8 | 300 | RQNSF | 4 | 4 | 2.0 | 1024 | 128 |
| | Models in Table 7 | | | | | | | | | |
| Model | Number of epochs | Training strategy | Batch size | Inference net hidden dim | NF type | Number of NF layers | Beam size | Length penalty | Max input tokens | Max target tokens |
| FlowSUM (Planar) | 3 | standard | 8 | 600 | Planar | 4 | 4 | 2.0 | 1024 | 128 |
| FlowSUM (Radial) | 3 | $\beta_C$-VAE | 8 | 600 | Radial | 4 | 4 | 2.0 | 1024 | 128 |
| FlowSUM (Sylvester) | 3 | $\beta_C$-VAE | 8 | 600 | Sylvester | 4 | 4 | 2.0 | 1024 | 128 |
| FlowSUM (RealNVP) | 3 | standard | 8 | 600 | RealNVP | 4 | 4 | 2.0 | 1024 | 128 |
| FlowSUM (IAF) | 3 | 1/3 CAAT[b] | 8 | 600 | IAF | 6 | 4 | 2.0 | 1024 | 128 |
| FlowSUM (RLNSF) | 3 | $\beta_C$-VAE | 8 | 600 | RLNSF | 4 | 4 | 2.0 | 1024 | 128 |
| FlowSUM (RQNSF) | 3 | 1/3 CAAT | 8 | 600 | RQNSF | 4 | 4 | 2.0 | 1024 | 128 |

[a] "-" means not applicable.
[b] 1/3 CAAT: aggressive training for 1 epoch and non-aggressive training for 2 epochs.

Table 11: Model Hyper-parameters.

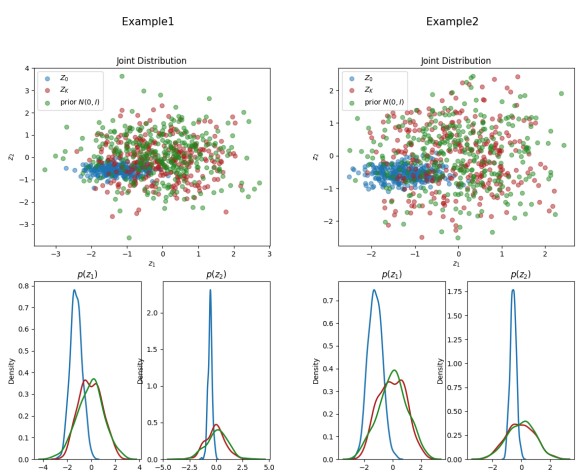

Figure 5: Visualization of the first two dimensions of $z_0$, $z_K$, and $N(0, I)$ by FlowSUM-PLKD on XSum. Both sub-figures demonstrate distinct multi-modal patterns.

## J  Normalizing Flows

**Planar flow** Proposed by Rezende and Mohamed (2015), the planar flow can be expressed as in Eq. 11. It applies contractions or expansions in the direction perpendicular to the hyperplane $\mathbf{w}^\top \mathbf{z} + b = 0$. Its Jacobian determinant can be computed in time $\mathcal{O}(D)$ as in Eq. 12, using the matrix determinant lemma. In addition, we need to note that this flow is not invertible for all values of $\mathbf{u}$ and $\mathbf{w}$. When the derivative of the activation function $h'(\cdot)$ is positive and bounded from above, $\mathbf{w}^\top \mathbf{u} > -\frac{1}{\sup_x h'(x)}$ is sufficient to ensure invertibility[15].

---

[15]In our code, we perform a transformation on $\mathbf{u}$ : $\mathbf{u} \leftarrow \mathbf{u} + \left[\log\left(1 + \exp\left(\mathbf{w}^\top \mathbf{u}\right)\right) - 1 - \mathbf{w}^\top \mathbf{u}\right] \cdot \frac{\mathbf{w}}{\mathbf{w}^\top \mathbf{w}}$ and restrict the activation $h(\cdot)$ to be one of leakyrelu, relu, and tanh to meet this condition.

$$f(\mathbf{z}) = \mathbf{z} + \mathbf{u}h\left(\mathbf{w}^\top \mathbf{z} + b\right), \qquad (11)$$

$$\det J = 1 + h'\left(\mathbf{w}^\top \mathbf{z} + b\right)\mathbf{w}^\top \mathbf{u} \qquad (12)$$

where $\{\mathbf{u}, \mathbf{w} \in \mathbb{R}^D, b \in \mathbb{R}\}$ are free parameters and $h(\cdot)$ is a smooth element-wise non-linear activation function with derivative $h'(\cdot)$.

**Radial flow** The radial flow (Tabak and Turner, 2013; Rezende and Mohamed, 2015) takes the form of Eq. 13. It applies radial contractions and expansions around a reference point. Similar to the planar flow, we can apply the matrix determinant lemma to calculate the Jacobian determinant in $\mathcal{O}(D)$ time, as in Eq. 14. To guarantee invertibility, we usually require $\beta > -\alpha^{16}$.

$$f(\mathbf{z}) = \mathbf{z} + \beta h(\alpha, r)(\mathbf{z} - \mathbf{z}_0), \qquad (13)$$

$$\det J = \left(1 + \frac{\alpha\beta}{h^2}\right)(1 + \beta h)^{D-1} \qquad (14)$$

where $\mathbf{z}_0 \in \mathbb{R}^D$ is the reference point, $\beta \in \mathbb{R}, \alpha \in \mathbb{R}^+$ are free parameters, $r = \|z - z_0\|$ is the norm of $z - z_0$, and $h(\alpha, r) = \frac{1}{\alpha + r}$.

**Sylvester flow** The Sylvester flows (van den Berg et al., 2018) generalize the planar flows to have $M$ hidden units, as in Eq. 15. To achieve better computational efficiency, van den Berg et al. (2018) proposes the parameterization as in Eq. 16, with which the Jacobian determinnant reduces to Eq. 17 and can be computed in $\mathcal{O}(M)$. Similar to the planar flows, when $h'(\cdot)$ is positive and bounded from above, $\tilde{\mathbf{R}}_{ii}\mathbf{R}_{ii} > -\frac{1}{\sup_x h'(x)}$ for all $i \in \{1, \ldots, D\}$ is sufficient to ensure invertibility.

$$f(\mathbf{z}) = \mathbf{z} + \mathbf{U}h\left(\mathbf{W}^\top \mathbf{z} + \mathbf{b}\right), \qquad (15)$$

where $\{\mathbf{U} \in \mathbb{R}^{D \times M}, \mathbf{W} \in \mathbb{R}^{D \times M}, \mathbf{b} \in \mathbb{R}^M\}$ are the free parameters and $h(\cdot)$ is an element-wise activation function.

$$f(\mathbf{z}) = \mathbf{z} + \mathbf{Q}\mathbf{R}h\left(\tilde{\mathbf{R}}\mathbf{Q}^T\mathbf{z} + \mathbf{b}\right), \qquad (16)$$

$$\det J = \det\left(\mathbf{I}_M + \text{diag}\left(h'\left(\tilde{\mathbf{R}}\mathbf{Q}^T\mathbf{z} + \mathbf{b}\right)\right)\tilde{\mathbf{R}}\mathbf{R}\right) \qquad (17)$$

where $\mathbf{R}$ and $\tilde{\mathbf{R}}$ are upper triangular $M \times M$ matrices, and $\mathbf{Q} = (\mathbf{q}_1 \ldots \mathbf{q}_M)$ consists of an orthonormal set of vectors.

[16]In our code, we perform a transformation on $\beta : \beta \leftarrow -\alpha + \log\left(1 + e^\beta\right)$ to guarantee invertibility.

**Autoregressive Flows** The masked autoregressive flow (MAF) (Papamakarios et al., 2017) was motivated by MADE (Germain et al., 2015), which is an autoregressive model for density estimation. MAF generalizes the conditional distribution to be Gaussian and generates data in a recursive way as in Eq. 18. Given a data point $\mathbf{x}$, the inverse transformation can be performed in parallel as in Eq. 19. The Jacobian of the inverse transformation is lower-triangular by design due to the autoregressive structure, hence its absolute determinant can be expressed as in Eq. 20. The set of functions $\{f_{\mu_i}, f_{\alpha_i}\}$ are autoregressive neural networks following the approaches in MADE.

$$x_i = u_i \exp \alpha_i + \mu_i, \qquad (18)$$

where $\mu_i = f_{\mu_i}(\mathbf{x}_{1:i-1}), \alpha_i = f_{\alpha_i}(\mathbf{x}_{1:i-1})$ and $u_i \sim \mathcal{N}(0, 1)$.

$$u_i = (x_i - \mu_i)\exp(-\alpha_i) \qquad (19)$$

$$\left|\det J^{-1}\right| = \exp\left(-\sum_i \alpha_i\right) \qquad (20)$$

Likewise, the inverse autoregressive flow (IAF) (Kingma et al., 2016) uses MADE with Gaussian conditionals and generates data as in Eq. 21. Its Jacobian determinant has a simple form as in Eq. 22. The main difference between IAF and MAF lies in the history variables. MAF uses previous data variables $\mathbf{x}_{1:i-1}$ to compute $\mu_i$ and $\alpha_i$, whereas IAF uses previous random variables $\mathbf{u}_{1:i-1}$ for the computation. In terms of sampling and density evaluation, IAF can sample in parallel and need to evaluate sequentially, whereas MAF has to sample sequentially and can evaluate in parallel. Since we care more about the sampling efficiency in variational inference, we choose IAF in the paper.

$$x_i = u_i \exp \alpha_i + \mu_i, \qquad (21)$$

where $\mu_i = f_{\mu_i}(\mathbf{u}_{1:i-1})$ and $\alpha_i = f_{\alpha_i}(\mathbf{u}_{1:i-1})$.

$$|\det J| = \exp\left(\sum_i \alpha_i\right) \qquad (22)$$

**Affine Coupling** The affine coupling layer, proposed in NICE (Dinh et al., 2015) and later generalized in RealNVP (Dinh et al., 2017) takes the following form.

$$\begin{cases} y_{1:d} &= x_{1:d} \\ y_{d+1:D} &= x_{d+1:D} \odot \exp\left(s\left(x_{1:d}\right)\right) + t\left(x_{1:d}\right) \end{cases}$$
$$(23)$$

where $s : R^d \mapsto R^{D-d}$ and $t : R^d \mapsto R^{D-d}$ are scale and translation transformation function respectively, and $\odot$ is the element-wise product.

Its Jacobian determinant can be efficiently computed as $\det J = \exp\left[\sum_j s\left(x_{1:d}\right)_j\right]$. Since the computation does not involve the Jacobian of $s$ or $t$, we can make these two functions arbitrarily complex and use neural networks to model them. The coupling layers are usually composed of permutation layers to ensure every component gets modified, and since the Jacobian determinant of permutation is 1, the Jacobian determinant remains tractable.

**Spline Coupling** Neural spline flows (Durkan et al., 2019; Dolatabadi et al., 2020) use monotonic rational-quadratic splines or monotonic rational-linear splines as the coupling transformation to achieve more flexibility and yet remain differentiable and invertible. The monotonic rational-quadratic spline uses $K + 1$ monotonically increasing knots $\left\{\left(x^{(k)}, y^{(k)}\right)\right\}_{k=0}^K$ to set up $K$ bins, each of which is defined as a rational-quadratic function[17] that is monotonically increasing. It maps $[-B, B]$ to $[-B, B]$ and defines the transformation outside the range to be identity transformation. Let $s_k = \left(y^{k+1} - y^k\right) / \left(x^{k+1} - x^k\right)$ and $\xi(x) = \left(x - x^k\right) / \left(x^{k+1} - x^k\right)$, the rational-quadratic function in the $k$th bin takes the form of Eq. 24 and the Jacobian determinant of the rational-quadratic neural spline flows (RQNSF) can be written as in Eq. 25.

$$\frac{\alpha^{(k)}(\xi)}{\beta^{(k)}(\xi)} = y^{(k)} + \frac{\left(y^{(k+1)} - y^{(k)}\right)\left[s^{(k)}\xi^2 + \delta^{(k)}\xi(1-\xi)\right]}{s^{(k)} + \left[\delta^{(k+1)} + \delta^{(k)} - 2s^{(k)}\right]\xi(1-\xi)}$$
$$(24)$$

$$\det J = \prod_k \frac{\mathrm{d}}{\mathrm{d}x}\left[\frac{\alpha^{(k)}(\xi)}{\beta^{(k)}(\xi)}\right]$$
$$= \prod_k \frac{\left(s^{(k)}\right)^2\left[\delta^{(k+1)}\xi^2 + 2s^{(k)}\xi(1-\xi) + \delta^{(k)}(1-\xi)^2\right]}{\left[s^{(k)} + \left[\delta^{(k+1)} + \delta^{(k)} - 2s^{(k)}\right]\xi(1-\xi)\right]^2}$$
$$(25)$$

The rational-linear neural spline flows (RLNSF) work similarly, except with monotonically increasing linear rational functions in each bin. Neural splines combine the best of autoregressive flows and coupling layers (such as NICE and RealNVP) in that it has both an analytic single-pass inverse and sufficient flexibility, as demonstrated in Durkan et al. (2019).

## K Example Analysis

In this section, we analyze several instances from CNN/Daily Mail and XSum, showcasing diverse outcomes generated by different summarization models.[18]

---

[17]A rational-quadratic function is defined as the quotient of two quadratic polynomial functions.

[18]It is worth mentioning that a few of the grammatical errors in the summaries can be attributed to the source text itself.

**Original Text (truncated):** It looks like an ordinary forest, with moss climbing up the walls and brown leaves covering the floor. But if you look closely, you will see that this picture is not all it seems. For the peaceful scene actually features a carefully painted female model. The amazing illusion is the work of German body-painting artist Joerg Duesterwald, who spent hours painting his model so she would blend in with her surroundings. The stunning set of pictures was taken in a forest in Langenfeld, Germany, yesterday. Mr Duesterwald has been painting for more than 20 years.

**Gold Summary:** The illusion is the work of German body-painting artist Joerg Duesterwald, who spent hours painting his model. Stunning set of pictures was taken in front of a rockface in a forest in Langenfeld, Germany, yesterday.

**BART:** Stunning set of images was taken in a forest near Langenfeld, Germany, yesterday by body-painting artist Joerg Duesterwald. It looks like an ordinary forest, with moss climbing up the walls and brown leaves covering the floor. But, if you look closely, you will see that this picture is not all it seems. For the peaceful scene actually features a carefully painted female model.

**VEDSUM:** The stunning set of pictures was taken in a forest in Langenfeld, Germany, yesterday. It looks like an ordinary forest, with moss climbing up the walls and brown leaves covering the floor. But, if you look closely, you will see that this picture is not all it seems. For the peaceful scene actually features a carefully painted female model.

**FlowSUM:** Amazing illusion is the work of German body-painting artist Joerg Duesterwald. He spent hours painting his model so she would blend in with surroundings. Stunning set of pictures was taken in a forest in Langenfeld, Germany, yesterday.

Table 12: Example 6766 in the CNN/Daily Mail test set: FlowSUM > BART > VEDSUM.

**Original Text (truncated):** UFC light heavyweight champion Jon Jones ran from a crash that hospitalised a pregnant woman - but quickly came back to grab 'a large handful of cash' from the car, witnesses told police. According to police, the accident occurred in southeastern Albuquerque just before noon on Sunday local time when the driver of a rented SUV jumped a red light. The driver, whom an off-duty officer identified as Jones, ran from the scene but then returned for the cash before fleeing again, police said. 'Witnesses stated he shoved the cash into his pants and ran north jumping the fence,' the report said. Officers found a pipe with marijuana in the vehicle as well as MMA and rental car documents in Jones' name, according to the police report. Police were searching for UFC champion Jon Jones in connection with a hit-and-run accident. Albuquerque police were seeking an arrest warrant for Jones on Monday. They said he would likely face a felony charge of leaving the scene of an accident since the woman broke her arm in the crash. Police said in a news release they'd been unable to reach Jones or his lawyer. However, Jones handed himself in later the same day, with TMZ reporting he was being held at Bernalillo County Metropolitan Detention Center. According to the warrant, the pregnant woman told police she was driving when she was hit by a silver Buick SUV. ··· Although he is widely considered the world's best pound-for-pound mixed martial artist, Jones has endured legal problems and questionable behaviour as champion.

**Gold Summary:** UFC light heavyweight champion Jon Jones ran from a crash that hospitalised a pregnant woman, witnesses told police. According to police, the accident occurred in Albuquerque just before noon on Sunday when the driver of a rented SUV jumped a red light. The driver, whom an off-duty officer identified as Jones, ran from the scene but then returned for the cash before fleeing again, police said. Jones is widely considered the best pound-for-pound mixed martial artist.

**BART:** Albuquerque police were seeking an arrest warrant for Jones on Monday. They said he would likely face a felony charge of leaving the scene of an accident since the woman broke her arm in the crash. However, Jones handed himself in later the same day, withTMZ reporting he was being held at Bernalillo County Metropolitan Detention Center.

**VEDSUM:** UFC light heavyweight champion Jon Jones ran from a crash that hospitalised a pregnant woman. Witnesses said he returned for 'a large handful of cash' from the car. Albuquerque police were seeking an arrest warrant for Jones on Monday. They said he would likely face a felony charge of leaving the scene of an accident since the woman broke her arm in the crash. Jones handed himself in later the same day.

**FlowSUM:** UFC light heavyweight champion Jon Jones ran from a crash that hospitalised a pregnant woman. Witnesses said he came back to grab 'a large handful of cash' from the car, witnesses told police. The driver, whom an off-duty officer identified as Jones, ran from the scene but then returned for the cash before fleeing again, police said. Officers found a pipe with marijuana in the vehicle as well as MMA and rental car documents in Jones' name, according to the police report.

Table 13: Example 4627 in the CNN/Daily Mail test set: FlowSUM > VEDSUM > BART.

**Original Text (truncated):** ... An Icelandic duo has created a snack that is made using cricket flour. Called the Jungle Bar it also contains dates, sesame seeds and chocolate. Cricket flour is said to be a good source of protein and other nutrients. The duo hopes it will encourage people in the West to eat more insects. The Jungle Bar is being developed by Icelandic duo Búi Bjarmar Aðalsteinsson and Stefán Atli Thoroddsen through their company Crowbar Protein. On Kickstarter they are seeking £10,000 ($15,000) for the insect-powered protein bar. They previously rose to fame with their Fly Factory, a micro-factory that used larvae to create foods including chocolate cake and pâté. Ingredients. Dates, sesame, sunflower and pumpkin seeds, chocolate and cricket flour. Nutrition information. The bar is 50 grams (1.7 ounces), 200 calories, contains 8 grams of high quality protein (16 per cent of the bar) and has a shelf life of 1 year. The duo say that insects are a largely untapped source of nutrients in the Western world, and they hope their product could spark a change in diet habits. To make the bar, the team has cricket flour sent in, from farm-raised crickets that have been ground down. It is then mixed with the other ingredients to make the unusual bar.

---

**Gold Summary**: Icelandic duo has created a snack that is made using cricket flour. Called the Jungle Bar it also contains dates, sesame seeds and chocolate. Cricket flour is said to be a good source of protein and other nutrients. The duo hopes it will encourage people in the West to eat more insects.

---

**BART**: An Icelandic duo has created a snack that is made using cricket flour. Called the Jungle Bar it also contains dates, sesame seeds and chocolate. Cricket flour is said to be a good source of protein and other nutrients. The duo hopes it will encourage people in the West to eat more insects. On Kickstarter they are seeking £10,000 ($15,000) for the insect-powered bar.

---

**VEDSUM**: The Jungle Bar is being developed by Icelandic duo Búi Bjarmar Aðalsteinsson and Stefán Atli Thoroddsen through their company Crowbar Protein. Cricket flour is said to be a good source of protein and other nutrients. The duo hopes it will encourage people in the West to eat more insects. To make the bar, the team has cricket flour sent in, from farm-raised crickets that have been ground down.

---

**FlowSUM**: An Icelandic duo has created a snack that is made using cricket flour. Called the Jungle Bar it also contains dates, sesame seeds and chocolate. The Jungle Bar is being developed by Búi Bjarmar Aðalsteinsson and Stefán Atli Thoroddsen through their company Crowbar Protein. On Kickstarter they are seeking £10,000 ($15,000) for the insect-powered protein bar. They previously rose to fame with their Fly Factory, a micro-factory that used larvae to create foods including chocolate cake and pâté.

Table 14: Example 4120 in the CNN/Daily Mail test set: BART > VEDSUM > FlowSUM.

**Original Text (truncated):** The Office for National Statistics said industrial output fell 0.7% compared with January, when it dropped 0.3%. Unexpectedly warm weather drove the change, because it led to a fall in electricity and gas demand, the ONS said. Construction output fell by 1.7% in February, down from a revised January reading of zero growth. The construction figure, the biggest drop in nearly a year, was mainly the result of a 2.6% fall in the housebuilding sector. Meanwhile, the UK's deficit in goods and services widened to Â£3.7bn in February, from a revised figure of Â£3bn in January. According to the ONS, the deficit was fuelled by what it called "erratic items", such as imports of gold and aircraft. "The overall trade deficit worsened, but excluding erratic items, the picture improved, as imports fell more than exports," said ONS senior statistician Kate Davies. Howard Archer, chief UK and European economist at IHS Markit, called the figures "a disappointing package of data for the UK economy which fuels suspicion that GDP growth slowed markedly, largely due to consumers becoming more cautious". He added: "We suspect UK GDP growth in the first quarter of 2017 slowed to 0.4% quarter-on-quarter from 0.7% quarter-on-quarter in the fourth quarter of 2016 - this would be the weakest growth rate since the first quarter of 2016."

**Gold Summary**: Activity in the UK's industrial and construction sectors shrank in February, new figures show.

**BART**: UK industrial output fell for the second month in a row in February, official figures have shown.

**VEDSUM**: Industrial output in the UK fell for the second month in a row in February, official figures have shown.

**FlowSUM**: Activity in the UK's industrial and construction sectors shrank in February, according to official figures.

Table 15: Example 2924 in the XSum test set: FlowSUM > BART > VEDSUM.

**Original Text (truncated):** In December, the government announced finalised plans for a cull, initially in pilot areas, as a way to curb the spread of tuberculosis in cattle. In applying for judicial review, the Badger Trust says culling will not stop TB and may in fact help spread it. Other campaign groups are considering action under the Bern Convention, which protects European wildlife. The government's plans are likely to result in farmers funding contractors to shoot badgers in a number of areas of England, with two initial pilots in west Gloucestershire and west Somerset taking place later this year. "We have identified some serious flaws in the way by which the Secretary of State [Caroline Spelman] reached her decision to cull badgers," said Gwendolen Morgan of Bindmans solicitors, lawyer for the Badger Trust. "Given that Defra's proposals come at an enormous cost to farmers, and threaten to prompt rather than prevent the spread of disease, we hope that this ill-conceived decision will be struck down by the court." She pointed to government projections that culling would reduce TB incidence by 12-16% over nine years.

**Gold Summary**: The Badger Trust has launched a new legal challenge to the government's plans to cull badgers in England.

**BART**: The Badger Trust has launched a legal challenge to the government's plans to cull badgers in England.

**VEDSUM**: The Badger Trust is taking legal action against the Department for Environment, Food and Rural Affairs (Defra) over plans to cull badgers in England.

**FlowSUM**: The Badger Trust has launched a legal challenge to the UK government's plans to cull badgers in England and Wales.

Table 16: Example 5737 in the XSum test set: BART > FlowSUM > VEDSUM.

**Original Text (truncated):** The response from many in that time has been: "Let's get on with it." That view was shared by the First Minister Carwyn Jones until recently when he altered his opinion and said that we should only start the official Brexit negotiations in the early part of next year. My sense is that the public will be flexible on the timing up to a point, as long as they are given a clear sense of direction. The majority of the political establishment have had to come to terms with the fact that most people ignored their advice to remain. So much for being in touch with the electorate. In conversations with politicians on the remain side since, I have come across a mix of bewilderment, frustration and sadness. And while people like me spend a lot of time talking and writing about a Welsh political dynamic, on this subject at least, Wales was a carbon copy of England. In stark contrast, those that supported leaving feel vindicated by their campaign, and now believe they are the ones in touch with vast swathes of the population. The referendum result was a devastating indictment of the effectiveness of the billions of pounds of EU funds spent trying to regenerate economically deprived communities. The brutal reality is that those who were most likely to vote to leave lived in communities where most EU money had been spent. It is an extraordinary paradox that raised eyebrows far further afield than Wales.

---

**Gold Summary**: It has been a month since Wales voted to leave the European Union.

---

**BART**: It has been more than a year since the UK voted to leave the European Union.

---

**VEDSUM**: It has been a year since the EU referendum result, and in that time I have spent a great deal of time talking to politicians on both sides of the political spectrum about what they think about Brexit.

---

**FlowSUM**: Since the referendum result on 23 June, I have spent a lot of time talking about the implications for Wales and the Welsh political establishment.

Table 17: Example 9512 in the XSum test set: BART > VEDSUM > FlowSUM.