# OpenReview forum: "Boosting Summarization with Normalizing Flows and Aggressive Training"
_EMNLP/2023/Conference — EMNLP 2023 Main_

### Official Review · Reviewer_BWR7 · 2023-08-04

**Typos Grammar Style And Presentation Improvements:** See Reasons To Reject.
**Soundness:** 3

**Excitement:**

3: Ambivalent: It has merits (e.g., it reports state-of-the-art results, the idea is nice), but there are key weaknesses (e.g., it describes incremental work), and it can significantly benefit from another round of revision. However, I won't object to accepting it if my co-reviewers champion it.

**Missing References:**

See Reasons To Reject.

**Paper Topic And Main Contributions:**

This paper proposed an NF-based VED framework with an aggressive training strategy for summarization tasks. Extensive experiments were conducted to reveal the effectiveness of the proposed method. However, there are still something need to be improved.

**Questions For The Authors:**

See Reasons To Reject.

**Reasons To Accept:**

1. This paper proposed an NF-based VED framework with an aggressive training strategy for summarization tasks.
2. Extensive experiments were conducted to reveal the effectiveness of the proposed method.

**Reasons To Reject:**

1. What are the potential meanings in Eq. (2) for ELBO? Corresponding references to support the backbone of the proposed framework are required.
2. Since the VED is the main framework in the proposed method, it is required to denote which parts in Figure 1 are variational posterior, variational prior, and encoder-decoder structure, respectively, making a better understanding.
3. Based on PLMs, the backbone of the proposed method, VED, seems could be directly applied for summarization tasks, how about its performance?
4. Unless NF and gate mechanism, what’s the main differences between the proposed VED and other VED frameworks? How about the performance of the model that introduces y into variational posterior instead of regrading p(z|x,y)=p(z|x)? Refer to reference: Variational Neural Machine Translation, EMNLP-2016.
5. Typos: (1) The paper said $six sections$ in line 127, but only 5 sections are presented; (2) Abbreviate forms should be defined when the corresponding phrases first appear and be capital letters, i.e., lines 163, 285, 1041.


**Reproducibility:**

4: Could mostly reproduce the results, but there may be some variation because of sample variance or minor variations in their interpretation of the protocol or method.

**Reviewer Confidence:**

4: Quite sure. I tried to check the important points carefully. It's unlikely, though conceivable, that I missed something that should affect my ratings.

---

> ### Author Rebuttal · Authors · 2023-08-26
>
> Dear Reviewer,
>
> Thank you for your comprehensive feedback.
>
> Regarding your first comment, the evidence lower bound (ELBO) is a surrogate for log-likelihood in parameter estimation within variational inference. Due to the intractability of $p(y | x)$, which prevents the direct use of maximum likelihood estimation for parameter estimation, we introduce a parametric distribution, the variational posterior $q(z | x, y)$, as an approximation to the true posterior $p(z | x, y)$.  As indicated in Appendix B, the KL divergence between these two distributions equates to the log-likelihood minus the ELBO. Hence, ELBO functions as a lower bound for log-likelihood, and the fidelity of $q(z | x, y)$ approximating $p(z | x, y)$ determines the efficacy of ELBO as a surrogate. We will update Section 2.2 to make the underlying idea clearer to the readers. Your observation regarding references for Variational Encoder-Decoders (VEDs) is duly noted. In addition to those cited in the Introduction, we plan to incorporate relevant references into Section 2.2, including Zhang et al., 2016; Serban et al., 2017; Zhou and Neubig, 2017; Shen et al., 2018, etc.
>
> To address your second comment, we intend to use distinct colors for the three key components. Specifically, the NF latent module, which models the variational posterior, will remain purple, the Transformer-based encoder-decoder (including $x_i$, Encoder Block, $x_i^{\prime}, y_j$, Decoder Block, $h_j$, and LM Head Layer) will be in green, and the refined gate mechanism (including Refined Gate and $h_j^{\prime}$) will be highlighted in orange. Furthermore, we will enhance the figure with detailed captions to clarify the model architecture.
>
> In response to your third comment, the backbone models are indeed applicable within the conventional VED framework. Notably, the integration of Pretrained Language Models (PLMs) within the traditional VED context aligns with a specific instance of FlowSUM, where the normalizing flow layer count $K = 0$.  Our earlier submission referenced this model as VAESUM and demonstrated its performance alongside FlowSUM. Given the underemphasized connection with FlowSUM, we plan to clarify this relationship in Section 3.1 and rename the model to VEDSUM for clarity.
>
> Regarding your fourth point, despite incorporating normalizing flows and the refined gate mechanism, the model architecture aligns with the traditional VED framework. Another difference lies in the proposed controlled alternate aggressive training (CAAT) strategy related to posterior collapse. Additionally, in terms of incorporating $y$ into the variational posterior, we choose to assume $q(z | x, y) = q(z | x)$ for the following reasons. Firstly, this assumption is grounded in the nature of summarization, where $y$ can be viewed as a condensed form of $x$ and hence it is sensible to assume all the information in $y$ is contained in $x$. Secondly, as evidenced by Zhang et al. 2016, it is plausible to condition the posterior on both $x$ and $y$. However, their approach suffers from difficulties during prediction. In prediction, the target text $y$ is not accessible, making it hard to sample from $q(z | x, y)$. Zhang et al. 2016 suggest taking the prior’s mean as the latent code, but in our paper, the prior is a Gaussian, and taking such a strategy would diminish the benefit of using normalizing flows. Thirdly, it has been shown empirically by Bryan Eikema & Wilker Aziz, 2018 that by restricting the conditioning of the posterior to $x$ alone, their model achieves higher accuracy. Therefore, we consider $q(z | x, y) = q(z | x)$ as our modeling strategy.
>
> Thank you for pointing out typographical errors and abbreviations in the manuscript. We are committed to addressing these issues and ensuring abbreviations are clearly defined. We highly appreciate your insightful feedback for enhancing our paper's clarity and quality.
>
> Best regards,
>
> ### References:
>
> Zhang, Biao, et al. "Variational Neural Machine Translation." *Proceedings of the 2016 Conference on Empirical Methods in Natural Language Processing*. 2016.
>
> Serban, Iulian, et al. "A hierarchical latent variable encoder-decoder model for generating dialogues." *Proceedings of the AAAI conference on artificial intelligence*. Vol. 31. No. 1. 2017.
>
> Zhou, Chunting, and Graham Neubig. "Morphological inflection generation with multi-space variational encoder-decoders." *Proceedings of the CoNLL SIGMORPHON 2017 Shared Task: Universal Morphological Reinflection*. 2017.
>
> Shen, Xiaoyu, et al. "Improving variational encoder-decoders in dialogue generation." *Proceedings of the AAAI conference on artificial intelligence*. Vol. 32. No. 1. 2018.
>
> Eikema, Bryan, and Wilker Aziz. "Auto-encoding variational neural machine translation." *arXiv preprint arXiv:1807.10564*(2018).

---

### Official Review · Reviewer_FZCB · 2023-08-05

**Soundness:** 4

**Excitement:**

4: Strong: This paper deepens the understanding of some phenomenon or lowers the barriers to an existing research direction.

**Paper Topic And Main Contributions:**

The paper introduces "FlowSUM," a novel variational encoder-decoder framework for Transformer-based summarization that addresses two key challenges: insufficient semantic information in latent representations and posterior collapse during training. To overcome these issues, the authors employ normalizing flows to enable flexible latent posterior modeling and propose a controlled alternate aggressive training (CAAT) strategy with an improved gate mechanism. Experimental results demonstrate that FlowSUM significantly improves the quality of generated summaries and facilitates knowledge distillation without a substantial increase in inference time.

**Reasons To Accept:**

1. The use of normalizing flows for variational summarization is a unique and promising idea, addressing important challenges in the field.
2. The introduction of a new training strategy with an improved gate mechanism is a valuable contribution to addressing posterior collapse during training.
3. The experiments are abundant.

**Reasons To Reject:**

1. Figure 1 is not clear which does not contain the NF latent module as described in the content.

**Reproducibility:**

3: Could reproduce the results with some difficulty. The settings of parameters are underspecified or subjectively determined; the training/evaluation data are not widely available.

**Reviewer Confidence:**

3: Pretty sure, but there's a chance I missed something. Although I have a good feel for this area in general, I did not carefully check the paper's details, e.g., the math, experimental design, or novelty.

---

> ### Author Rebuttal · Authors · 2023-08-26
>
> Dear reviewer,
>
> Thank you for pointing out the clarity concerns related to the figure. In the submitted manuscript, the NF latent module is colored in purple, encompassing the chain from $\overline{x}$ to $z_K$. To address your concern, we intend to enhance the figure's clarity by assigning distinct colors to the three crucial components: the NF latent module in purple, the Transformer-based encoder-decoder (including $x_i$, Encoder Block, $x_i^{\prime}, y_j$, Decoder Block, $h_j$, and LM Head Layer) in green, and the refined gate mechanism (comprising Refined Gate and $h_j^{\prime}$) in orange. Additionally, we'll include clear figure captions to provide further clarity on the model architecture. We genuinely appreciate your diligence in reviewing our content and are dedicated to delivering a more refined version based on your insights.
>
> Best regards,

---

### Official Review · Reviewer_6vj4 · 2023-08-07

**Typos Grammar Style And Presentation Improvements:** didn't notice any
**Soundness:** 4

**Excitement:**

4: Strong: This paper deepens the understanding of some phenomenon or lowers the barriers to an existing research direction.

**Missing References:**

https://arxiv.org/abs/1809.05233 seem to use VAEs for summarization but in a different way

**Paper Topic And Main Contributions:**

The authors introduce FlowSUM, a transformer-based encoder-to-decoder summarization framework that utilizes normalizing flows. They show the competitiveness to state-of-the-art summarization models and superiority over direct baselines. They also show the applicability of their model to knowledge destillation

**Questions For The Authors:**

none

**Reasons To Accept:**

- great motivation, well outlined
- well written paper
- extensive experiments on multiple datasets
- good ablation studies

**Reasons To Reject:**

none

**Reproducibility:**

3: Could reproduce the results with some difficulty. The settings of parameters are underspecified or subjectively determined; the training/evaluation data are not widely available.

**Reviewer Confidence:**

2: Willing to defend my evaluation, but it is fairly likely that I missed some details, didn't understand some central points, or can't be sure about the novelty of the work.

---

> ### Author Rebuttal · Authors · 2023-08-26
>
> Dear reviewer,
>
> Thank you for drawing attention to the overlooked reference. We highly value your thorough review of our manuscript. The reference you highlighted sheds light on an alternative approach to variational summarization—one that emphasizes unsupervised learning and explicit control over summary length using VAEs. This perspective represents a growing avenue in variational summarization, gaining momentum alongside the expanding availability of diverse large training corpus. Thank you for your recommendations. We will incorporate your invaluable feedback to enhance our work.
>
> Best regards,

---

### Meta-Review · Area_Chair_RmAB · 2023-09-14

**Recommendation:** 4

**Metareview:**

The paper introduces "FlowSUM," a novel variational encoder-decoder framework for Transformer-based summarization that solves two major problems: posterior collapse during training and a lack of semantic information in latent representations. To address these problems, the authors suggest a controlled alternate aggressive training (CAAT) technique with a better gate mechanism and use normalizing flows to enable flexible latent posterior modeling. According to experimental findings, FlowSUM greatly enhances the quality of generated summaries and speeds up knowledge distillation without dramatically lengthening inference times.

Pros:
1. The paper is well-written
2. The approach is novel
3. Extensive experiments that show that the proposed methodology works fine
4. Good ablation study

The authors addressed all comments (cons) that were raised by the reviewers in rebuttal.

---

### Decision · Program_Chairs · 2023-10-07

**Decision:**

Accept-Main

**Comment:**

The paper introduces "FlowSUM," a novel variational encoder-decoder framework for Transformer-based summarization that solves two major problems: posterior collapse during training and a lack of semantic information in latent representations. To address these problems, the authors suggest a controlled alternate aggressive training (CAAT) technique with a better gate mechanism and use normalizing flows to enable flexible latent posterior modeling. According to experimental findings, FlowSUM greatly enhances the quality of generated summaries and speeds up knowledge distillation without dramatically lengthening inference times.

Pros:
1. The paper is well-written
2. The approach is novel
3. Extensive experiments that show that the proposed methodology works fine
4. Good ablation study

The authors addressed all comments (cons) that were raised by the reviewers in rebuttal.